# Relationship between Phenolic Compounds and Antioxidant Activity of Some Moroccan Date Palm Fruit Varieties (*Phoenix dactylifera* L.): A Two-Year Study

**DOI:** 10.3390/plants13081119

**Published:** 2024-04-17

**Authors:** Abdoussadeq Ouamnina, Abderrahim Alahyane, Imane Elateri, Abderrahim Boutasknit, Mohamed Abderrazik

**Affiliations:** 1Laboratory of Agro-Food, Biotechnologies and Valorization of Plant Bioresources (AGROBIOVAL), Departement of Biology, Faculty of Science Semlalia, Cadi Ayyad University, Marrakech 40000, Morocco; ouamnina@gmail.com (A.O.);; 2Agrobiotechnology and Bioengineering Center, CNRST-labeled Research Unit (AgroBiotech-URL-CNRST-05 Center), Cadi Ayyad University, Marrakech 40000, Morocco; 3Higher Institute of Nursing Professions and Health Techniques of Guelmim (ISPITSG), Guelmin 81000, Morocco; 4Multidisciplinary Faculty of Nador, Mohamed First University, Nador 62700, Morocco

**Keywords:** date palm fruit, harvesting season, phenolic compounds, valorization

## Abstract

In Morocco, the abundance of low-value varieties in the oases may provide an opportunity to capitalize on this richness to create new nutraceutical food products. In this context, the phenolic profile and antioxidant capacity of four Moroccan date varieties were analyzed. Our results indicate that the levels of total polyphenols, total flavonoids and total condensed tannins vary, respectively, from 91.86 to 364.35 mg GAE/100 g of dry weight (DW), 46.59 to 111.80 mg QE/100 g DW and 16.10 to 42.03 mg CE/100 g DW during the 2021 harvest season. Furthermore, during the 2022 harvest season, these contents vary, respectively, from 119.13 to 410.39 mg GAE/100 g DW, 59.30 to 110.85 mg QE/100 g DW and 21.93 to 53.95 mg CE/100 g DW. The results of the HPLC-UV-VIS analysis revealed that, in all four varieties, gallic acid was and remained one of the major compounds in the date extracts. In addition, a high antioxidant activity of date extracts was particularly observed in the three tests, namely ferric reducing power (FRAP), ferrous ion chelating capacity (FIC) and the phosphomolybdate test. This richness in phenolic compounds makes low-value dates a source of active ingredient that can replace the synthetic antioxidants used in the food and pharmaceutical industries.

## 1. Introduction

The date palm, scientifically known as *Phoenix dactylifera* L., was one of the first plants to be domesticated by human beings. Its fame extends beyond borders, particularly in the arid regions of the Middle East and North Africa, where more than 5000 date palm cultivars have been recorded worldwide [1,2,3]. Today, as a result of climate change, many regions are gradually adopting climatic conditions specific to date palm cultivation, characterized by long, hot summers, low rainfall and low relative humidity. This farming practice spread rapidly to other parts of the world [4], leading to an anticipated increase in global date production. Dates play a major role in the diets of many countries, particularly in Arab regions such as Saudi Arabia, Iran, Egypt and Algeria [5]. Dates are also processed and incorporated into various processed food products such as bread, cakes, biscuits, ice cream, chocolate bars, cereals, pasta, powders, jams, jellies, juices, syrups and vinegars [6]. In addition, this fruit is an excellent source of essential phenolic compounds [7].

The phytochemical content of dates is responsible for their antioxidant capacity thanks to their abundance of carotenoids, polyphenols, particularly phenolic acids such as gallic acid, ferulic acid, syringic acid, caffeic acid and coumaric acid, as well as isoflavones, lignans, flavonoids and tannins [7,8]. These compounds have the ability to neutralize free radicals. This ability is attributed to conjugated cyclic structures and hydroxyl groups, which act by scavenging superoxide anion, singlet oxygen and lipid peroxyl radicals [9]. Various factors such as climatic, agronomic and genomic conditions, pre- and post-harvest stages and processing can significantly influence the phenolic composition of the fruit [10].

Phenolic compounds, which are secondary metabolites, have various roles, including protection against different diseases [11]. Epidemiological research has shown that consuming foods rich in polyphenols could be beneficial in preventing various conditions, including asthma and cancer [12]. In addition, they may help prevent diseases such as atherosclerosis, diabetes and neurodegenerative conditions, including Parkinson’s and Alzheimer’s disease [13,14]. They are also associated with a reduction in inflammation and have the potential to prevent certain cardiovascular diseases [15,16]. On the other hand, phenolic compounds are widely exploited in industry for a diverse range of uses. For example, gallic acid, one of the predominant compounds found in dates [17], has a wide range of industrial applications, notably as a food additive and cosmetic ingredient. Its ester derivatives are widely incorporated into processed foods and food packaging materials to prevent oxidative rancidity and spoilage. Gallic acid is frequently used to stabilize collagen during the leather manufacturing process. It is also used as a raw material in the production of inks, paints, color developers and pharmaceutical products [18].

Morocco is renowned for its abundant date palm cultivation, which thrives in the southern Atlas region. This area includes valleys such as the Drâa, Ziz, Gheris, Toudgha and Figuig, containing around 2.8 million palm trees in 453 varieties [19]. These varieties generated a production of 150,301 tons in 2021 [20]. Despite this rich Moroccan heritage, the sector faces a number of challenges. Firstly, these oases are under increasing pressure from human activities on natural resources that have already been depleted due to the prolonged effects of drought, water shortages and the “Bayoud” disease [21]. The notable shortcomings in the marketing of the Moroccan date palm product are mainly related to the poor characterization of Moroccan date varieties. The most marketable date varieties are Majhoul, Boufeggous, Jihel and Bouskri [22]. Our study will therefore focus on a rarely studied variety (Khalt Khal) and two varieties that have never been studied before (Rasse Tmar and Jdar Lahmer).

Four varieties were studied in this context. Three of them have a low market value (the rarely studied Khalt Khal, as well as Jdar Lahmer and Rasse Tmar, two varieties that have never been studied). These varieties are very abundant in Moroccan oases, while a fourth variety, Majhoul, has a high market value. The aim of the study was to assess changes in polyphenols in the different date varieties over two harvest seasons (2021 and 2022) and to analyze their contribution to antioxidant power. By highlighting the underestimated value of these varieties, we aim to contribute to the valorization of these varieties with a low market value by proposing ways of valorization such as the use of these varieties in the pharmaceutical industry or the creation of new products with a high market value. We also aim to promote the conservation and sustainable use of these varieties in Moroccan oases, which could play a decisive role in preserving our country’s botanical heritage, while stimulating economic opportunities within oasis communities.

## 2. Results

### 2.1. Total Phenolic Compounds, Total Flavonoids and Total Condensed Tannins

The content of total phenols, total flavonoids and total condensed tannins determined by spectrophotometry for varieties harvested in 2021 and 2022 is presented in Figure 1.

Significant differences (*p* < 0.001) in total polyphenol content were found between the date varieties studied. Total polyphenol contents ranged from 91.86 to 364.35 mg GAE/100 g dry weight in 2021 and from 119.13 to 410.39 mg GAE/100 g dry weight in 2022 (Figure 1a). In both harvest seasons, the Khalt Khel variety had the highest total phenolic compound content, followed by Mjhoul, Rasse Tmar and Jdar Lahmer. The levels of phenolic compounds determined by the Folin–Ciocalteu method were generally higher than those determined by HPLC. This difference is probably due to the presence of other reducing agents, such as reducing sugars, in date extracts [23].

Total flavonoid content ranged from 46.59 to 111.80 mg QE/100 g DW and from 59.30 to 100.85 QE/100 g DW in 2021 and 2022, respectively (Figure 1c). For both harvest seasons, the Jdar Lahmer variety had the lowest total flavonoid content. This difference in flavonoid content was significant (*p* < 0.001) compared with the other three varieties, Khalt Khal, Rasse Tmar and Majhoul, which showed no significant difference between them.

Significant differences were observed in the TCT (total condensed tannin) content between the four date varieties harvested in two seasons (*p* < 0.001, 2021; *p* < 0.001, 2022). This content varied from 16.10 to 42.03 mg CE/100 g DW in 2021 and from 21.93 to 53.95 mg CE/100 g DW in 2022 (Figure 1b). Based on this TCT content, the varieties can be classified into two groups. The first group includes the two varieties Khalt Khal and Jdar Lahmer, which show no significant difference but have a high TCT content in 2021 and 2022. The second group, characterized by low TCT content, includes Jdar Lahmer and Majhoul, for which no significant difference was observed. However, there was a significant difference between the two groups (*p* < 0.001).

The variation in the content of total phenolic compounds, total flavonoids and total condensed tannins for the same cultivars during the two seasons was not statistically significant. Possibly, this is because the four varieties studied—Khalt Khal, Jdar Lahmer Majhoul and RasseT mar—are either stable in their phytochemical compositions or the climatic conditions during the two study years 2021 and 2022 are almost similar.

### 2.2. Antioxidant Activity

Various in vitro techniques have been developed to assess the antioxidant efficacy of fruit. However, there is still no standardized protocol on the method to be used. In this study, the antioxidant capacity of date palm fruit varieties was assessed using four separate tests: (1) DPPH (2,2-diphenyl-1-pierylhydrazyl) radical scavenging assay. This test consists of measuring the decrease in absorbance of the violet chromogenic radical 2,2-diphenyl-1-picrylhydrazyl (DPPH-) reduced by antioxidant compounds to pale-yellow hydrazine, at its absorption maximum at 517 nm [24]. (2) The ferric reducing power test is based on the reduction of the Fe³+/ferricyanide complex to the ferrous form in the presence of antioxidant substances in the antioxidant samples. The complex formed (blue coloration) can be measured at 700 nm [25]. (3) The phosphomolybdate test is based on the reduction of molybdenum (IV) to molybdenum (V) by the analyzed sample, with the formation of green phosphate/Mo (V) compounds showing maximum absorption at 695 nm [26]. (4) The ferrous ion chelating (Fe^2+^) assay is measured by inhibiting the formation of the Fe^2+^–ferrozine complex by the absorbance of this complex at 562 nm [27]. The antioxidant power expressed by the 50% inhibition concentration of date palm fruit extracts is shown in Table 1.

The DPPH test is expressed by IC_50_ in mg/mL (the concentration of extract required or positive control to inhibit 50% of initial DPPH). The antioxidant power of date fruit extracts tested by DPPH ranged from 13.19 to 54.83 mg/mL in 2021 and in 2022 from 15.41 to 41.80 mg/mL. The varieties with the higher antioxidant activity for the two harvest seasons were Khalt khal and Majhoul (no significant difference between these two varieties between the two harvest years for the same variety as well) followed by Jdar lahmer and Rasse tmar (no significant difference between the two harvest years for the same variety, but the difference between these two varieties is significant in 2021). The DPPH test values for the four varieties in 2021 and 2022 are significantly different from the positive controls used *p* < 0.001 (Trolox 0.15 mg/mL; BHT 0.75 mg/mL).

In 2021, the phosphomolybdenum test values (Table 1) for the cultivars varied from 0.49 mg/mL in the Khalt Khal variety to 0.61 mg/mL in the Majhoul variety. However, the difference between genotypes was not significant (*p* > 0.05). On the other hand, in 2022, the values of this test ranged from 0.57 mg/mL in the Khalt Khal variety to 0.87 mg/mL in the Majhoul variety, and the difference between varieties was significant (*p* < 0.001), except between Jdar Lahmer and Rasse Tmar, which were significantly similar (*p* > 0.05). When we compare the date fruit harvest seasons, we notice that the antioxidant activity tested by phosphomolybden decreased in 2022. Three genotypes showed significantly different values between the two harvest seasons (Jdar Lahmer, Rasse Tmar and Majhoul). In contrast, the Khalt Khal variety showed similar values in both harvest seasons. The phosphomolybdenum test values for the four varieties in 2021 and 2022 were significantly different from the positive controls used *p* < 0.001 (Trolox 0.28 mg/mL; BHT 0.23 mg/mL).

The mean antioxidant activity values of dates determined by the FIC test are shown in Table 1. The Khalt Khal, Rasse Tmar and Majhoul varieties showed high antioxidant activity, comparable to that of the EDTA positive control (0.15 mg/mL) in 2021, while they were significantly different from the Jdar Lahmer variety, which showed the lowest antioxidant activity. In 2022, the antioxidant activities of the dates ranged from 0.08 mg/mL to 1.36 mg/mL. The Khalt Khal variety showed extremely high antioxidant activity compared to Jdar Lahmer and Rasse Tmar, and comparable to the positive control (EDTA). On the other hand, the Jdar Lahmer variety always had the lowest antioxidant activity. However, the variation in antioxidant activity of the varieties between the two years was not significant (*p* > 0.05).

FRAP tests carried out to evaluate the antioxidant activity of different date cultivars revealed significant differences, particularly between Khalt Khal, Jdar Lahmer and Rasse Tmar with the Majhoul variety (*p* < 0.001) (Table 1). In addition, a significant difference (*p* < 0.001) in antioxidant activity was observed between the two years, but only for the Majhoul variety. The FRAP test also determined that the antioxidant activity of date extracts varied from 1.71 to 3.46 mg/mL in 2021 and from 2.11 to 4.51 mg/mL in 2022 (Table 1). In comparison, the antioxidant activity of date extracts was significantly different from the positive control used in this test, Trolox, which has a high antioxidant power of 0.11 mg/mL. This difference can be explained by the purity of Trolox compared with date extracts, which contain a mixture of molecules. It is interesting to note that the higher-value varieties (Khalt Khal, Jdar Lahmer and Rasse Tmar) had twice the antioxidant power of the Majhoul variety, which is considered to have a high market value.

### 2.3. Phenolic Compound Profiles by HPLC-UV-VIS Detector

Analysis of the phenolic compounds present in the four date varieties by HPLC is presented in Table 2. A total of 12 phenolic compounds were identified and quantified by comparing their retention times and UV spectra with those of standards analyzed under similar conditions. These compounds include gallic acid, tyrosol, *trans*-ferulic acid, p-hydroxyphenylacetic acid, caffeic acid, vanillic acid, ellagic acid, epicatechin, catechin, quercetin, vanillin and kaempferol.

Firstly, the analysis revealed that tyrosol and quercetin were not detected in all varieties, in either the 2021 or 2022 samples. With regard to the major compounds in 2021, in the Khalt Khal variety, the three most important compounds were vanillic acid (16.11 mg/100 g DW), followed by ellagic acid (14.62 mg/100 g DW) and finally gallic acid (9.26 mg/100 g DW). For the Jdar Lahmer variety, the main compounds are gallic acid (15.55 mg/100 g DW), vanillic acid (9.56 mg/100 g DW) and ellagic acid (5.34 mg/100 g DW). In the case of the Rasse Tmar variety, the principal compounds are the gallic acid (15.86 mg/100 g DW), the ellagic acid (12.23 mg/100 g DW) and the p-hydroxyphenylacetic acid (4.59 mg/100 g DW). Finally, in the Majhoul variety, the major compounds are gallic acid (10.83 mg/100 g DW), followed by p-hydroxyphenylacetic acid (6.89 mg/100 g DW) and vanillic acid (6.43 mg/100 g DW).

For the 2022 harvest season, phenolic compound levels follow a similar trend to 2021. In the Khalt Khal variety, the three principal compounds are ellagic acid, followed by vanillic acid and finally gallic acid. In the Jdar Lahmer variety, gallic acid, vanillic acid and ellagic acid are the principal compounds. In the Rasse Tmar variety, the major compounds are gallic acid, ellagic acid and p-hydroxyphenylacetic acid. Finally, in the Majhoul variety, the main compounds are gallic acid, vanillic acid and ellagic acid.

The different varieties showed significant variation in terms of phenolic compounds detected by HPLC (*p* < 0.001). This disparity is probably due to the genotype, which is also valid for the divergences observed within the same varieties between the two harvest seasons (2021 and 2022). This variation is also significant (*p* < 0.001), thus confirming the impact of the harvest season on phenolic compound content. For gallic acid, the difference between the two harvest seasons was only significant for the Khlat Khal and Jdar Lahmer varieties. As for *trans*-ferulic acid and caffeic acid, the difference is observed in the Rasse Tmar variety between the two seasons. As for p-hydroxyphenylacetic acid, the difference was noted in the Majhoul variety. And for vanillic acid, ellagic acid, catechin and vanillin, the significant difference between the two seasons was detected in the Khalt Khal variety. Finally, no significant difference was detected between the samples harvested during the two seasons for the four varieties concerning epicatechin and kaempferol.

### 2.4. Correlation

In order to evaluate the influence of various elements such as TPs, TFAs, TCTs, and various compounds detected by high-performance liquid chromatography (HPLC) on the antioxidant activity of date pulp extracts, we calculated Pearson correlation coefficients (Figure 2). A strong negative correlation was found between the IC_50_ values of the DPPH test and the antioxidant component contents (total phenols, r = −0.83; total flavonoids, r = −0.52; catechin, r = −0.45; vanillic acid, r = −0.47; *trans*-ferulic acid, r = −0. 49; *p* < 0.05), as well as between the IC_50_ values of the FIC test and the content of antioxidant components (total phenols, r = −0.8; total flavonoids, r = −0.95; condensed tannins, r = −0.51; p-hydroxyphenylacetic acid, r = −0.73; caffeic acid, r = −0.44; epicatechin and vanillin, r = −0.43; *p* < 0.05). A significant negative correlation (*p* < 0.05) was also observed between the IC_50_ values of the FRAP test and gallic acid (r = −0.55), ellagic acid (r = −0.48), epicatechin (r = −0.46) and vanillic acid (r = −0.42). Finally, a strong negative correlation (*p* < 0.05) between the phosphomolybdenum test measured by IC_50_ and phenolic compounds was observed in our results (caffeic acid, r = −0.81; epicatechin, r = −0.64; catechin, r = −0.57; vanillic acid, r = −0.55; vanillin, r = −0.47; ellagic acid, r = −0.42). The correlations observed between phenolic compounds and antioxidant activity confirm the existing link between these two parameters. In other words, the antioxidant activity of date extracts is associated with the quantity of phenolic compounds present. 

This analysis also makes it possible to visualize the compounds implicated in each test, which could explain the high activity of some varieties compared to others. For example, the Khalt Khal variety shows particularly high activity compared to other varieties in the four antioxidant tests. This intense activity is attributable to its high phenolic compound content (Khalt Khal variety), which is responsible for each test.

### 2.5. Principal Component Analysis (PCA) and Cluster Analysis

Principal component analysis (PCA) is a statistical analysis widely used to evaluate the correlation between various variables and to identify groupings within the sample under study. In this study, PCA was applied to a group of 17 variables collected from four date varieties harvested in two regions for two successive seasons.

Principal component analysis conducted on the four varieties revealed that 68.19% of the total variance was explained by the first two principal components (Figure 3a,b). The first component (PC1) explained 46.40% of the total variability, while the second (PC2) explained 21.97%. This variability explained by the first and second components is created by the contribution of a number of variables. For example, the first five variables contributing to the first component (PC1) are vanillin (11.10%), catechin (10.96%), epicatechin (10.58%), ellagic acid (10.32%) and vanillic acid (9.18%). On the other hand, the first five variables contributing to the second component (PC2) are DPPH (19.72%), FRAP (17.86%), PT (12.63%), FT (9.53%) and kaempferol (8.47%). The five variables contributing to PC1 are positively correlated with the first component, while the five variables contributing to PC2 are negatively correlated with the second component, with the exception of DPPH and kaempferol, which are positively correlated with this component (Table 3).

Negative correlations were observed between antioxidant activity and total phenolics, total flavonoids, total condensed tannins and all compounds detected by HPLC. This is to be expected, as IC_50_ values are inversely proportional to antioxidant potential (Figure 3 and Table 3).

The distribution of varieties (Figure 3a) according to the variables revealed considerable variability between them. The ‘Rasse Tmar’ and ‘Jdar Lahmer’ varieties were positively correlated with DPPH and FIC antioxidant activities, as well as with phytochemical variables (gallic acid, caffeic acid and kaempferol), but were negatively correlated with the FRAP and the phosphomolybdenum tests. On the other hand, the ‘Majhoul’ variety correlated negatively with DPPH and FIC antioxidant activities, but positively with FRAP, the phosphomolybdenum test, PT, FT and *trans*-ferulic acid. The ‘Khalt Khal’ variety clearly stood out from the other varieties. Principal component analysis (PCA) revealed that this distinction is mainly due to the abundance of phytochemical compounds (catechin, epicatechin, vanillin and vanillic acid) in this variety. The variety’s strong positive correlation with these compounds explains its high antioxidant activity.

Furthermore, in Figure 3a, all the varieties are distributed in distinct locations, with the exception of the points that are very close together, which denote dates of the same variety but harvested in two seasons. We can therefore conclude that the variability between these four varieties is mainly attributable to the genotype, and that the effect of the seasons on the variability is minimal.

To better visualize this variability among the varieties, a hierarchical clustering analysis (HCM) was conducted (Figure 4a). Three groups were generated by this analysis, with each group containing the same varieties but harvested in two different seasons, except for the third group, which includes the two varieties Jdar Lahmer and Rasse Tmar. This analysis confirms our conclusion that the variability among the varieties is mainly due to genotype, while the effect of the season on variability is weak. Furthermore, we can also conclude that the two varieties Jdar Lahmer and Rasse Tmar exhibit some similarity between them. Finally, to unambiguously determine the characteristics of each group, which will impact the choice of valorization route for each variety, an analysis of parallel coordinates was conducted (Figure 4b).

According to the parallel coordinates analysis (Figure 4b), the first group comprises the variety Khalt Khal, characterized by high antioxidant activity (DPPH test, FRAP, FIC, phosphomolybdenum) and high levels of phytochemical compounds (TPs, FTs, CTCs, compounds analyzed by HPLC), except for kaempferol. The second group includes the variety Majhoul, characterized by high antioxidant activity (DPPH and FIC), as well as high total polyphenol and total flavonoid contents, with low levels of phenolic compounds analyzed by HPLC, except for p-hydroxyphenylacetic acid and trans-ferulic acid, according to the parallel coordinates plot. In contrast, the third group comprises the two varieties Jdar Lahmer and Rasse Tmar, characterized by high antioxidant activity only in relation to the FRAP test, and low levels of phytochemical compounds compared to the first group, according to the parallel coordinates plot.

## 3. Discussion

Phenolic compounds are one of the most extensively researched classes of bioactive compounds due to their well-documented positive effects on health. They are recognized for their ability to act as antioxidants thanks to their capacity to provide a hydrogen atom and/or an electron to free radicals. The antioxidant effect exercised by these compounds depends closely on the number and arrangement of the hydroxyl groups they contain [28,29]. However, these compounds have a number of notable characteristics, including anti-inflammatory, antimicrobial and anti-proliferative properties [28,30]. Another scientific study showed that the association of quercetin with iron oxide nanoparticles administered to rats led to a significant improvement in learning capacity and memory in these animals [31]. These biological activities have generated considerable interest in the potential use of these molecules in the creation of nutraceuticals [28,29]. For example, phenolic extract from *Rosmarinus officinalis* L. is used as an additive in fromage frais because of its proven antioxidant properties [32]. In addition, anthocyanins are used in a variety of food products such as apple juice, sports drinks, yoghurt, syrups and sweets, where they play an essential role in stabilizing color, enriching the product with bioactive molecules, improving color, antioxidant and antimicrobial activities and sensory characteristics [33,34,35,36].

According to previous studies, our results show a similarity with reports indicating that the total content of phenolic compounds in methanolic extracts varied from 101.06 to 478.37 mg GAE/100 g DW in seventeen Moroccan varieties [7], and varied in aqueous extracts from 127.97 to 334.58 mg GAE/100 g DW in eight Algerian varieties tested [37], and in aqueous extracts from 171.39 to 353.92 mg GAE/100 g DW in five Moroccan date varieties [38]. However, our results show high levels compared with those reported by Zineb et al. [39], which vary from 41.80 to 84.73 mg GAE/100 g DW. On the other hand, our results are lower than those of Pakkish and Mohammadrezakhani [40], who found total polyphenol contents ranging from 82.47 to 98.31 mg GAE/g DW in seven Iranian varieties. The variations observed between the different varieties could be explained by the fact that the phenolic compound content of the fruit can be influenced by various factors such as genotype, cultivation practices, geographical origin, ripening stage, harvesting time and storage time [14,41].

Our results are similar with the results obtained by the authors of [17,38], who found that the total flavonoids in an aqueous extract ranged from 43.28 to 84.956 mg of quercetin equivalent/100 g DW in five varieties of Moroccan dates, and from 19.62 to 299.74 mg of quercetin equivalent/100 g DW in acetone (60%) extracts of ten varieties of Algerian dates, respectively. However, the results found in our study are higher than the values reported in several studies [40,42], which showed that the total flavonoid content varied from 31 to 45 mg quercetin equivalent/100 g DW and from 1.06 to 4.23 mg catechin equivalent/100 g DW, respectively. These observed differences may be due to growing conditions, agricultural practices, storage conditions, climatic factors, ripening stage, solubility and solvent extractability of flavonoids [3,43,44,45]. However, the health benefits of flavonoids are widely recognized, including their anti-inflammatory and anti-cancer properties, their ability to neutralize free radicals and chelate metals, as well as their role in reducing chronic diseases and preventing cardiovascular disorders [46].

Tannins are responsible for the astringency of unripe dates. As the dates ripen, the tannins start to precipitate out, causing them to lose their astringency. In addition, tannins play a role in the development of the color of dates after they have been harvested [47]. 

Tannin levels in our studies are lower than those found by El arema et al. [48] and Bouhlali et al. [49], which ranged from 54.93 to 102.37 mg CE/100 g DW in five Tunisian varieties in Tamar stage and 57.56 to 92.14 mg CE/100 g DW in eight Moroccan varieties, respectively. However, our results fall within the ranges of values found in 17 Moroccan cultivars and five Algerian varieties [7,50]; the condensed tannin contents in methanolic extracts vary from 5.29 to 152.15 mg CE/100 g EC and from 3.58 to 60.05 mg CE/100 g EC, respectively. An earlier study of two Saudi date varieties (Khudari and Sullaj) found very low levels of tannins in the Tamar stage, ranging from 1.33% to 1.58% [51]. However, it is essential to stress that variations between varieties are influenced by the use of different solvents and extraction methods, environmental conditions, harvesting season, as well as cultivars and ripening stages [48,52,53].

For the study of antioxidant activity of date palm fruits, three Tunisian and five Algerian date varieties were evaluated for their ability to chelate ferrous ions. The results of this study revealed a low antioxidant activity compared with our own values, with IC_50_ values ranging from 85.19 to 91.71 mg/mL, and from 16.36 to 771 mg/mL, respectively [50,51,52,53,54]. On the other hand, Sudanese date fruits exhibited antioxidant activity relatively consistent with our own results. This activity ranged from 59.33% to 79.78% for a concentration of methanolic extract of 0.5 mg/mL [55]. In addition, the phosphomolybdenum [56] revealed high antioxidant activity in three varieties harvested in Saudi Arabia, with IC_50_ values ranging from 12.30 to 192.66 µg/mL, which is close to our results [56].

However, the results of the antioxidant activity determined by the DPPH test obtained in another study [54] are in agreement with our results, with IC_50_ values ranging from 16.70 to 46.79 mg/mL of acetone/H_2_O extract (70:30) for the three Tunisian varieties studied.

An FRAP study carried out by the authors of [48] revealed that Tunisian dates showed high antioxidant activity compared to our results, with IC_50_ values ranging from 0.08 to 0.24 mg/mL. Another study conducted on methanolic extracts [7] obtained IC_50_ results ranging from 0.219 to 2.028 mg/mL for seventeen Moroccan varieties, which is consistent with our findings. In contrast, the antioxidant activity of two Saudi Arabian date varieties, Tamazouchete and Takarboucht, was lower compared to our results, with IC_50_ values ranging from 10.19 and 25.68 mg/mL, as reported by the authors of [50].

This difference in antioxidant activity between the different varieties could be the result of disparities in cultivation methods and climatic conditions between the different sites, particularly variations in temperature, soil moisture constraints and the availability of mineral nutrients. Soil characteristics and fertilization also influence the nutritional composition and antioxidant capacity of harvested fruit [14]. Another study cited an effect of foliar applications such as boron on date palm quality [57].

The antioxidant activity of phenolic compounds is mainly due to their redox properties, which enable them to act as reducing agents, hydrogen donors and singlet oxygen inhibitors. They may also have metal chelating potential [58]. In general, for polyphenols, the presence of at least one phenolic OH group makes these compounds active. So, the number and position of hydroxyl groups in the aromatic ring, and the methoxy substituents in the *ortho* position relative to the OH are of crucial importance in modulating antioxidant capacity [9,59,60]. In addition, flavonoids have an antioxidant effect, but this activity varies from one molecule to another depending on the degree of hydroxylation and the position of the –OH groups in the B ring. In particular, an *ortho*-dihydroxylated structure of the B ring results in higher activity because it confers more stability on the aroxyl radical through electron delocalization [61] or acts as a preferred binding site for trace metals [62].

One study [63] examined the use of phenolic compounds of plant origin as natural antioxidants in various edible oils. In addition, another study [64] suggested that phenolic compounds from grapes could be as effective as propyl gallate in preventing the oxidation of fish oil emulsions in water. Another research has shown that phenolic compounds extracted from mango kernel powder extended the shelf life of buffalo ghee [65]. In addition, methanolic extracts from wild rice hulls have been found to inhibit lipid oxidation in minced meat [66].

The results obtained from the literature concerning the various phenolic compounds analyzed by HPLC are as follows. According to Al Harthi et al. [67], the gallic acid content in the fruits of different date palm varieties harvested in Oman varies from 7.0 mg/100 g to 19.14 mg/100 g. Similarly, the level of caffeic acid varies from 0.34 mg/100 g to 1.75 mg/100 g, and that of vanillic acid ranges from 0.18 mg/100 g to 0.27 mg/100 g. According to the observations of Al Juhaimi et al. [68] on dates harvested in Saudi Arabia, the principal phenolic constituents are gallic acid (between 1.61 mg/100 g and 11.23 mg/100 g), (+)-catechin (between 0.29 mg/100 g and 2.98 mg/100 g), benzene-1, 2-diol (between 0.57 mg/100 g and 2.88 mg/100 g), syringic acid (between 0.08 mg/100 g and 1.36 mg/100 g), 3,4-dihydroxybenzoic acid (between 0.3 mg/100 g and 2.68 mg/100 g) and caffeic acid (between 0.13 mg/100 g and 1.37 mg/100 g). In addition, the study conducted on methanolic extracts by the authors of [7] indicates that Moroccan dates are particularly rich in gallic acid, followed by catechin, with quantities varying, respectively, from 5.568 mg/100 g to 31.411 mg/100 g for the first compound and from 1.867 mg/100 g to 7.275 mg/100 g for the second. These different studies show that gallic acid remains one of the major compounds in dates, which confirms our own results.

However, El Sohaimy et al. [69] found that the ethanolic extract of an Egyptian variety had an extremely low concentration of gallic acid, not exceeding 0.528 mg/100 g dry matter. Furthermore, Khallouki et al. [70] reported very limited levels of kaempferol and quercetin, namely 1.28 mg/kg and 1.01 mg/kg, respectively, in date palm fruit extract which is different from our results. The variation in phenolic compound levels in dates of different varieties can be attributed to several factors, including environmental conditions, climate, temperature, humidity, cultivar, ripening stage, harvesting season, geographical origin, fertilizer application, soil type, exposure to sunlight, storage conditions, extraction solvent, extraction conditions, among others [14,43,71,72,73,74].

Date extracts show various activities, including anti-diabetic, antimicrobial, anti-tumoral, anti-inflammatory and antioxidant effects [71,75,76,77,78]. The presence of phenolic compounds in dates is mainly responsible for their antioxidant activity. The structure of these compounds also plays a key role in their antioxidant potential. For example, benzoic acids with an –OH position *ortho*- or *para*- to –COOH show no antioxidant effect [60]. Another study shows that the antioxidant activity of phenolic acids increases with the degree of hydroxylation. Trihydroxylated gallic acid, in particular, shows strong antioxidant activity. However, substitution of the hydroxyl groups in positions 3 and 5 with methoxyl groups reduces its antioxidant effectiveness [60].

Other studies indicate that hydroxycinnamic acids have a higher antioxidant activity than hydroxybenzoic acids, possibly due to the CH=CH–COOH group, which confers a stronger capacity to donate H and neutralize radicals than the –COOH group of hydroxybenzoic acids [60,79]. In the case of flavonoids, the nature of the substitutions on the B and C rings is decisive for their antioxidant activity [80].

The results of our correlation study of total polyphenols, total flavonoids and total condensed tannins are consistent with previous findings. Alahyane et al. [7] identified negative correlations of these compounds with two antioxidant assays, namely FRAP (ferric reducing antioxidant power) and DPPH (2,2-diphenyl-1-picrylhydrazyl). Similar observations were reported by Amorós et al. [81], who found a strong correlation between phenolic content and overall antioxidant activity during the Khalal stage of seven palm varieties. Similarly, Mansouri et al. [71] found a significant correlation between phenolic compound content and the anti-free radical effectiveness of seven varieties of ripe Algerian dates (r^2^ = 0.975). These conclusions are supported by the findings of [17], which observed significant correlations between phenolic compounds and all three antioxidant tests (FRAP, DPPH and ferrous ion chelating capacity), which is in agreement with our current results. In addition, observations reveal a negative correlation between antioxidant activity measured by the DPPH and FRAP assays and the presence of vanillic acid (r = −0.241 and −0.141), as well as an inverse correlation between catechin (r = −0.181) and gallic acid (r = −0.309) with the FRAP assay, and between vanillin (r = −0.211) and the DPPH assay [7]. These results are consistent with the findings of our current study. Furthermore, a study conducted by Žilić et al. [82] reported a negative correlation between some phenolic acids, notably ferulic acid, and antioxidant capacity measured by the DPPH assay in durum wheat.

However, some studies have reported the absence of correlation between the total concentration of phenolic compounds and the antioxidant activity of date extracts [67]. This observation can be explained by the specific nature of the phenolic compounds extracted by the extraction solvent, which confirms that these extracts, as well as polyphenols, do not represent the only phytochemical compounds responsible for the antioxidant activity of date fruits. 

Various factors could explain the variations in the correlation between the bioactive compounds and antioxidant activity, such as compromising mechanisms of specific-action antioxidants in each test [83]. These mechanisms include the ability of phenolic compounds to attenuate oxidative stress by electron or hydrogen transfer processes, as well as their ability to capture metal ions or prevent lipid peroxidation [9].

The results of our study are consistent with those of [7], which examined the correlation of variables in 17 varieties. Their research showed that total phenolics, total flavonoids, condensed tannins and HPLC-identified compounds can influence variability between different varieties. The results of the principal component analysis Farag et al. [84] indicate that the distribution of flavones and flavonols in the fruit is the principal element of distinction between cultivars. The same study mentioned that the most distinct cultivars in the cluster analyses showed the strongest antioxidant effect. These phenolic compounds could be used as molecular markers to assess diversity between date cultivars and to identify priority cultivars for development initiatives and valorization [85].

## 4. Materials and Methods

### 4.1. Reagents and Chemicals

Sodium carbonate, Folin–Ciocalteu phenol reagent, ferrozine, sodium nitrite, aluminum chloride, sodium hydroxide, hydrochloric acid, vanillin, catechin, gallic acid and quercetin, potassium chloride, phosphate buffer, potassium ferricyanide, trichloroacetic acid, phosphomolybdate reagent, 6-hydroxy-2,5,7,8-tetramethylchroman-2-carboxylic acid, ethylene diamine tetra acetic acid (eDTA), fructose, 1,1-diphenyl-2- picrylhydrazyl (dPPH), bovine serum albumin (BSA); HPLC standards.

### 4.2. Plant Material

The fruits (*Phoenix dactylifera* L.) were harvested at the Tamar stage during two consecutive seasons (23 October 2021 and 23 October 2022) in two regions, Zagora and Erfoud. These regions, located in Morocco, benefit from an arid and semi-arid climate, thus offering optimal climatic conditions for this study.

Four varieties were considered: Khalt Khal, Jdar Lahmer and Majhoul, harvested in the Zagora region, and Rasse Tmar, harvested in the Erfoud region. The coordinates of the palm trees selected in the sampling for each variety are shown in Table 4.

The diagonal sampling method was used to target the palm trees to be sampled for each variety. In the field, by arranging the sampling points diagonally, a fixed distance of 10 m was maintained over all the study sites. Consistent tree selection assured uniformity in tree length and spacing across the various sites. Dates were harvested from different plants of the same variety, from different clusters. Two kilograms of fruits was harvested from 10 randomly selected trees per replication, with a total of three replications. 

After harvesting, immediate steps were taken to maintain the quality of the fruit during transport to the laboratory. The samples were carefully stored in a cool box to preserve their integrity. On arrival at the food science laboratory at the Semlalia Faculty of Science in Marrakech, Cadi Ayyad University, the harvested fruit was quickly stored at −20 °C. This rapid preservation method was crucial for safeguarding the phenolic compounds and preventing any degradation [86].

### 4.3. Preparation of Ethanolic Extract

Extraction of phenolic compounds was conducted using a protocol with slight adjustments [69]. A total of 5 g (×3) of pitted date fruit (*Phoenix dactylifera* L.) was ground and macerated at 25 °C for 4 h under magnetic stirring with 50 mL of 60% ethanol. Subsequently, the mixture was centrifuged at 4000 rpm for 20 min. The supernatant was collected, and the same process was repeated three times on the resulting pellet. The final supernatant was then evaporated under reduced pressure, and the extract was collected and stored at −20 °C. The choice of a 60% ethanol was based on its established efficacy in extracting phenolic compounds [87].

### 4.4. Total Phenol (TP) Content Estimation

The method described by Singleton et al. [88] was employed, with minor adjustments, to quantify phenolic compounds. In this procedure, 0.1 mL of the diluted extract was combined with 3.9 mL of deionized water and 0.1 mL of Folin–Ciocalteu reagent. Three minutes later, 1 mL of sodium carbonate (7.5%) was introduced into the solution. The solution was then incubated at room temperature for 2 h in the dark and measured at 725 nm using a spectrophotometer. Gallic acid was used as a reference standard and the results were reported in milligrams of gallic acid equivalents per 100 g of dry matter (mg GAE/100 g DW).

### 4.5. Total Flavonoid Content (TFA) Estimation

The method employed by Zhishen et al. [89] with minor adjustments was used to determine the total flavonoid content. A total volume of 0.2 mL of extract was diluted with 1 mL of distilled water, and then 60 μL of sodium nitrite NaNO_2_ (5%) and 60 μL of aluminum chloride AlCl_3_ (10%) were added. Five minutes later, 0.4 mL of sodium hydroxide NaOH (1 M) was added to the mixture and the absorbance was immediately read at 510 nm. The results were reported as mg QE/100 g DW using quercetin as the standard.

### 4.6. Total Condensed Tannin (TCT) Content Estimation

The method used to estimate procyanidin levels is based on Heimler et al.’s work [90] with minor modifications. To conduct this, 0.2 mL of diluted sample was added to 1.5 mL of a vanillin solution (4% in methanol) and 0.75 mL of concentrated hydrochloric acid. The solution was incubated for 20 min before the absorbance was measured at 500 nm. The total amount of condensed tannins is expressed as mg catechin equivalent per 100 g dry weight (mg CE/100 g DW).

### 4.7. Quantification of Phenolic Compound by HPLC-UV-VIS Detector

Phenolic compounds were evaluated using high-performance liquid chromatography (SHIMADZU), equipped with a C18 column (250 nm × 4.6 mm) maintained at 30 °C and a UV/VIS PDA, SPD, M20A detector. The mobile phase was 5% formic acid (Solvent A) and methanol (Solvent B). The eluent gradient used was as follows: 0 min, 5% B; 55 min 100% B; 55–60 min, 100% B. The flow rate was 1 mL/min, the injection volume was 20 microliters, the detection length was 280 nm and the analysis time was 60 min. The identification of each peak was determined using the retention time of the standards used [91,92].

### 4.8. Antioxidant Activity

#### 4.8.1. DPPH (2,2-diphenyl-1-pierylhydrazyl) Radical Scavenging Assay

The method described by Mansouri et al. [71] was used to estimate antioxidant capacity. A total volume of 25 μL of the extracts at different concentrations was mixed with 0.975 mL of a methanolic solution of DPPH (6 × 10^−5^ M), and the solution was kept in the dark at 25 °C for 30 min. Absorbance was then determined at 517 nm. Trolox and BHT were used as positive controls. The percentage of inhibition (%I) was measured according to the equation below:%I = [(DO_control_ − DO_sample_)/DO_control_] × 100

DPPH free radical scavenging activity is expressed in terms of percentage of inhibition at 50% (IC_50_ mg/mL).

#### 4.8.2. FRAP Assay (Ferric Reducing Antioxidant Power)

The procedure of Oyaizu [93] was used to evaluate the ferric reducing power of date extracts. Extracts at different concentrations or Trolox (0.2 mL) were added to 0.5 mL of phosphate buffer (0.2 M, pH 6.6) and 0.5 mL of potassium ferricyanide (1%). The solution was incubated at 50 °C for 20 min, and then 0.5 mL of trichloroacetic acid solution (10%) was added. After combining 0.5 mL of the added mixture with 0.5 mL of distilled water and 0.1 mL of ferric chloride FeCl_3_ solution (0.1%), absorbance was determined at 700 nm.

#### 4.8.3. Antioxidant Capacity by the Phosphomolybdenum Assay

The protocol used by Prieto et al. [94] involved measuring the reducing power of the samples studied. To perform this, 0.3 mL of an extract at various concentrations or of a positive control (Trolox and BHT) were added to 3 mL of phosphomolybdate reagent (0.6 M sulfuric acid, 28 mM sodium phosphate and 4 mM ammonium molybdate), then kept at 95 °C for 90 min in the dark. After cooling, absorbance was measured at 695 nm. The antioxidant capacity of the samples is expressed as IC_50_ (mg/mL).

#### 4.8.4. Ferrous ion Chelating Assay (FIC)

According to Liyana-Pathirana and Shahidi [95], the chelating capacity of plant extracts was assessed using a method based on the inhibition of Fe (II)–Ferrozine complex formation. We then combined 0.4 mL of plant extract or a standard chelator (EDTA), 0.285 mL of distilled water and 0.275 mL of FeC_l2_.4H_2_O (0.2 mM). Five minutes after incubation, 40 μL of ferrozine (5 mM) was added to the mixture, which was then shaken and incubated for 10 min. Absorbance was determined at 562 nm. The chelating activity is represented as a percentage by the following equation:Chelating activity (%) = [(DO_control_ – DO_sample_)/DO_control_] × 100

The chelating capacity of ferrous ions is expressed as IC_50_ (mg/mL).

### 4.9. Statistical Analysis

Results were expressed as the mean of three replicates ± SD. Statistical analyses were performed using the R software version (R i386 4.0.5).

## 5. Conclusions

Studies conducted on the three date varieties with low market value showed that their contents of total phenolic compounds, total flavonoids and total condensed tannins were high or comparable to those of the variety with high market value (Majhoul). Analysis of the phenolic profile by HPLC/DAD revealed the presence of ten phenolic compounds. Gallic acid remains one of the predominant compounds in all date varieties. These bioactive compounds show significant correlations with the antioxidant activity of the varieties examined, particularly those with a low market value, which show a high antioxidant capacity. This study confirms that the differences between these date varieties are mainly attributable to genetic inheritance, while the impact of the seasons on this variation is minimal. This confers on these varieties a certain stability in terms of bioactive compounds during the seasons. In conclusion, this study has shown that low-market-value date varieties (Khalt Khal, Jdar Lahmer and Rasse Tmar) are a significant source of natural antioxidant compounds. These compounds could have potential applications in the pharmaceutical field and could also be considered as ingredients for functional foods, or used as food additives for various specific technological functions such as texture improvement, preservation, coloring, nutritional enrichment and flavoring.

## Figures and Tables

**Figure 1 plants-13-01119-f001:**
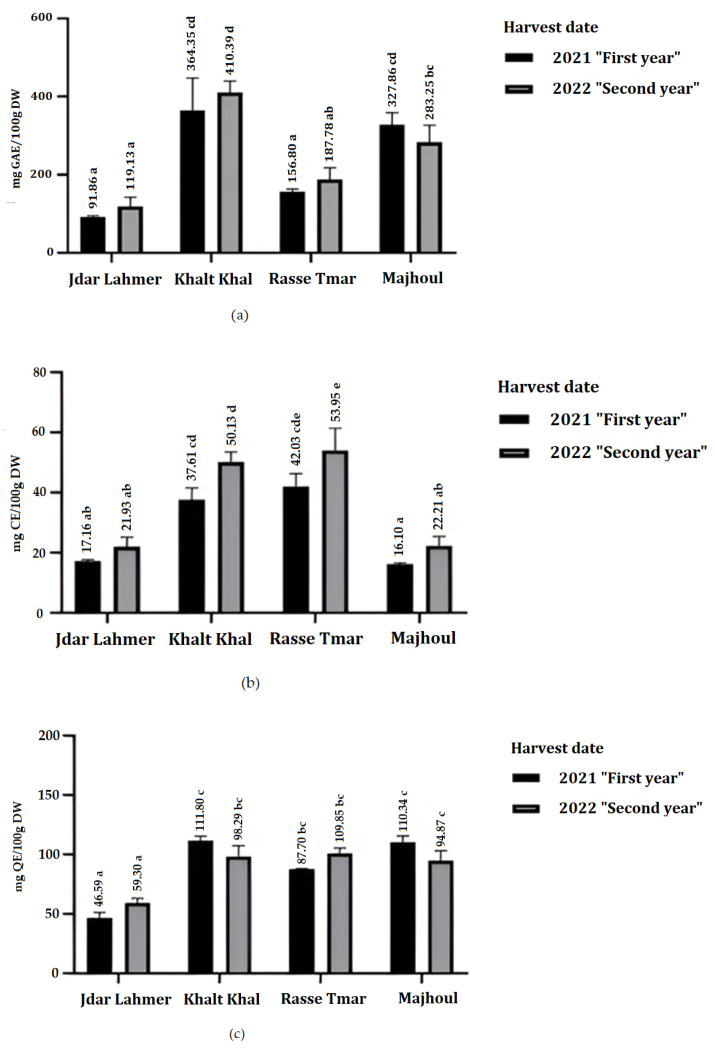
(**a**) Total phenolics (TPs) in mg GAE/100 g DW, (**b**) total condensed tannins (TCTs) contents in mg CE/100 g DW, and (**c**) total flavonoids (TFAs) in mg QE/100 g DW of various date palm fruit varieties harvested successively over two years. Means sharing the same letters are not significantly different at *p* < 0.05, as determined by Turkey’s test.

**Figure 2 plants-13-01119-f002:**
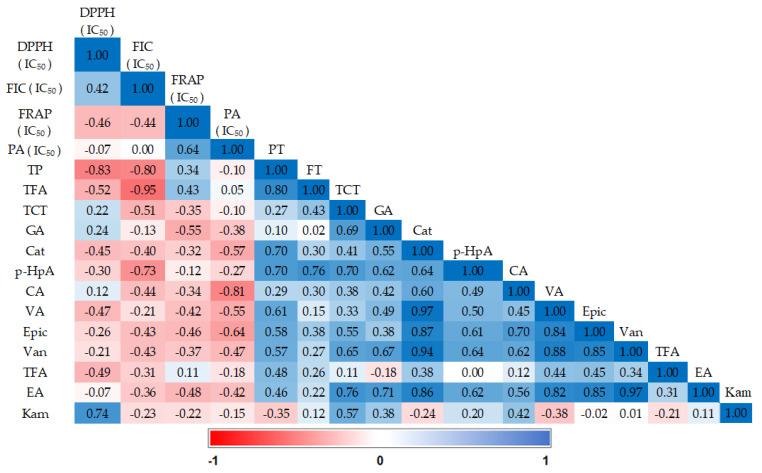
Pearson correlation coefficients between the variables of the date varieties studied; (2,2-diphenyl-1-picrylhydrazyl) radical scavenging assay (DPPH); ferrous ion chelating assay (FIC); ferric reducing antioxidant power assay (FRAP); phosphomolybden assay (PA); total phenolics (TP); total flavonoids (TFA); total condensed tannins contents (TCT); gallic acid (GA); catechin (cat); p-hydroxy phenylacetic acid (p-HpA); caffeic acid (CA); vanillic acid (VA; epicatechin (epic); vanillin (van); *trans*-ferulic acid (TFA); ellagic acid (EA) ; kaempferol (Kam).

**Figure 3 plants-13-01119-f003:**
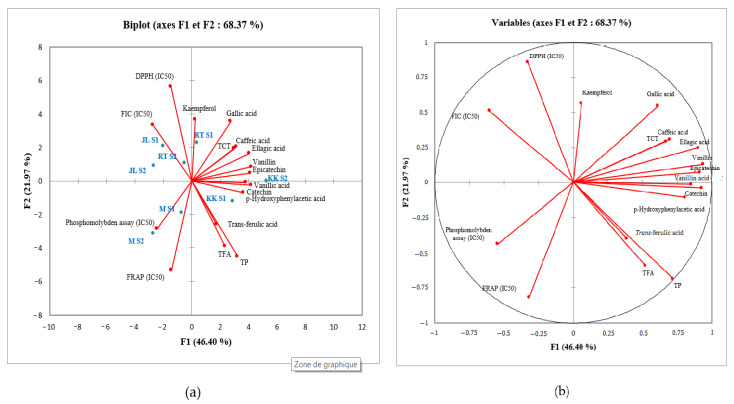
Biplot obtained from PCA of variables and individuals. (**a**) Segregation of four date varieties harvested in two seasons according to their phenolic compounds and antioxidant activity, (**b**) PCA correlation circle of studied parameters; KK: Khalt Khal; JL: Jdar Lahmer; RT: Rasse Tmar; M: Majhoul; S1: Harvest season 2021; S2: Harvest season 2022.

**Figure 4 plants-13-01119-f004:**
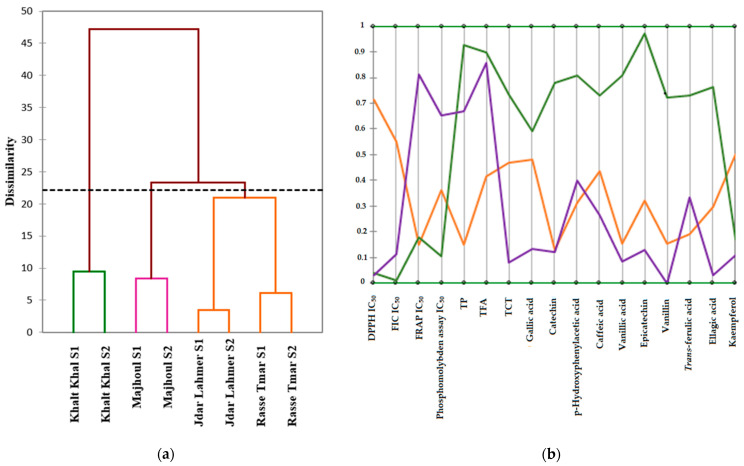
(**a**) Cluster dendrogram of the grouping of date varieties on the basis of the variables studied (HCM). (**b**) Parallel coordinate graph showing the characteristics of the three groups generated by HCM; S1: Harvest season 2021; S2: Harvest season 2022.

**Table 1 plants-13-01119-t001:** Antioxidant activity, expressed as IC_50_ values, for Moroccan date palm fruit varieties harvested successively over two years.

Harvest Date	Variety	DPPH (IC_50_ in mg/mL)	Phosphomolybden (IC_50_ in mg/mL)	FIC (IC_50_ in mg/mL)	FRAP (IC_50_ in mg/mL)
First year 2021	Khalt Khal	14.22 ± 0.08 ab	0.49 ± 0.008 b	0.05 ± 0.02 a	2.24 ± 0.11 bc
	Jdar Lahmer	37.36 ± 2.66 c	0.51 ± 0.009 b	1.58 ± 0.14 c	1.71 ± 0.23 b
	Rasse Tmar	54.83 ± 15.99 d	0.54 ± 0.01 b	0.21 ± 0.02 ab	2.48 ± 0.29 c
	Majhoul	13.19 ± 0.18 ab	0.61 ± 0.02 bc	0.19 ± 0.05 ab	3.46 ± 0.23 d
Second year 2022	Khalt Khal	15.41 ± 1.69 ab	0.57 ± 0.03 b	0.08 ± 0.02 a	2.17 ± 0.20 bc
	Jdar Lahmer	37.63 ± 0.86 c	0.71 ± 0.02 c	1.36 ± 0.24 c	2.11 ± 0.13 bc
	Rasse Tmar	41.80 ± 4.14 cd	0.74 ± 0.13 c	0.41 ± 0.05 b	2.18 ± 0.09 bc
	Majhoul	15.76 ± 1.52 b	0.87 ± 0.06 d	0.26 ± 0.05 ab	4.51 ± 0.29 e
Positive control	Trolox	0.15 ± 0.01 e	0.28 ± 0.01 a	-	0.11 ± 0.01 a
	BHT	0.75 ± 0.04 e	0.23 ± 0.01 a	-	-
	EDTA	-	-	0.155 ± 0.02 ab	-

Each value in the table is the mean ± standard deviation (*n* = 3). Letters (a–e) indicate significant differences at *p* < 0.001; BHT, butylated hydroxytoluene; EDTA, ethylenediaminetetraacetic acid; IC_50_, half-maximal inhibitory concentration.

**Table 2 plants-13-01119-t002:** Content of individual phenolic compounds determined by HPLC-DAD of four Moroccan date varieties harvested in two successive seasons (mg/100 g DW).

	Khalt Khal		Jdar Lahmer		Rasse Tmar		Majhoul	
	2021	2022	2021	2022	2021	2022	2021	2022
Gallic acid	9.26 ± 1.55 a	23.62 ± 0.05 d	15.55 ± 0.16 b	8.29 ± 0.02 a	15.86 ± 0.11 bc	18.46 ± 0.53 c	10.83 ± 3.98 ab	6.14 ± 0.09 a
*Trans*-ferulic acid	2.67 ± 0.11 f	1.42 ± 0.03 d	1.29 ± 0.01 d	0.39 ± 0.04 ab	0.77 ± 0.01 c	0.64 ± 0.01 bc	0.33 ± 0.17 a	1.88 ± 0.04 e
p-Hydroxyphenyl-acetic acid	6.09 ± 1.74 bc	9.41 ± 0.61 c	0.83 ± 0.05 a	1.12 ± 0.75 a	4.59 ± 0.01 ab	7.48 ± 0.09 bc	6.89 ± 2.40 bc	1.62 ± 0.37 a
Caffeic acid	2.25 ± 1.24 bc	2.17 ± 0.27 abc	1.28 ± 0.23 abc	0.81 ± 0.09 ab	2.89 ± 0.01 c	0.88 ± 0.01 ab	1.71 ±0.14 abc	0.38 ± 0.11 a
Vanillic acid	16.11 ± 3.39 c	24.36 ± 0.42 d	9.56 ±0.71 b	7.08 ± 2.03 ab	4.35 ± 0.23 ab	3.38 ± 0.21 a	6.43 ± 1.10 ab	2.78 ± 0.60 a
Ellagic acid	14.62 ± 5.40 d	27.32 ± 2.86 e	5.34 ± 1.76 abc	6.54 ± 0.52 a–d	12.23 ± 0.23 cd	9.75 ± 0.23 bcd	0.44 ± 0.12 a	2.13 ± 0.02 ab
Epicatechin	5.58 ± 1.34 c	5.29 ± 0.71 c	1.15 ± 0.11 ab	2.06 ± 0.04 ab	2.81 ± 0.33 b	1.60 ± 0.03 ab	1.55 ± 0.40 ab	0.17 ± 0.02 a
Catechin	5.37 ± 1.57 b	9.27 ± 0.53 c	2.11 ± 0.36 a	1.28 ± 0.62 a	2.31 ± 0.71 ab	0.60 ± 0.03 a	2.59 ± 1.13 ab	0.43 ±0.20 a
Vanillin	1.32 ± 0.21 c	2.98 ± 0.68 d	0.26 ± 0.05 ab	0.20 ± 0.04 ab	1.04 ± 0.01 bc	0.34 ± 0.01 abc	nd	nd
Kaempferol	0.96 ± 0.10 a	0.51 ± 0.22 a	0.98 ± 0.43 a	0.16 ± 0.05 a	3.52 ± 0.73 b	2.72 ± 0.15 b	0.54 ± 0.04 a	0.50 ± 0.14 a

Each value in the table is the mean ± standard deviation (*n* = 3). Letters (a–f) indicate significant differences at *p* < 0.001; 2021 and 2022: Harvest date; nd: not detected.

**Table 3 plants-13-01119-t003:** Correlations and contributions of variables in the two factors (F1 and F2).

	Correlations: Variables, Factors		Contribution of Variables (%)	
	F1	F2	F1	F2
DPPH (IC_50_)	−0.32	0.85	1.36	19.72
FIC (IC_50_)	−0.60	0.51	4.70	6.99
FRAP (IC_50_)	−0.31	−0.81	1.28	17.86
Phosphomolybden assay (IC_50_)	−0.54	−0.43	3.80	5.09
TPs	0.71	−0.68	6.55	12.63
TFAs	0.52	−0.59	3.44	9.53
TCTs	0.66	0.29	5.63	2.33
Gallic acid	0.61	0.54	4.73	7.91
Catechin	0.93	−0.04	10.96	0.04
p-Hydroxyphenylacetic acid	0.80	−0.10	8.24	0.31
Caffeic acid	0.69	0.30	6.12	2.55
Vanillic acid	0.85	−0.01	9.18	0.01
Epicatechin	0.91	0.07	10.58	0.13
Vanillin	0.93	0.13	11.10	0.45
*Trans*-ferulic acid	0.38	−0.39	1.90	4.23
Ellagic acid	0.90	0.25	10.32	1.69
Kaempferol	0.05	0.56	0.04	8.47

**Table 4 plants-13-01119-t004:** The coordinates of the palm sampling site during the two harvesting seasons.

Varieties	Origin	Latitude	Longitude	Altitude
Khalt Khal	Zagora	30.341944 N 30°20′30.98508″	−5.857113 W 5°49′37.84116″	727 m
Jdar Lahmer	Zagora	30.341829 N 30°20′30.52824″	−5.828390 W 5°49′42.12408″	727 m
Majhoul	Zagora	30.356288 N 30°21′22.63392″	−5.825682 W 5°49′32.45592″	727 m
Rasse Tmar	Erfoud	31.453796 N 31°27′13.66524″	−4.346465 W 4°20′47.29884″	805 m

## Data Availability

Data are contained within the article.

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
