# Peer review of "Relationship between Phenolic Compounds and Antioxidant Activity of Some Moroccan Date Palm Fruit Varieties (Phoenix dactylifera L.): A Two-Year Study"

_plants, 2024, doi:10.3390/plants13081119_

Round 1

Reviewer 1 Report

Comments and Suggestions for Authors

In this study, the authors investigated the composition and content of phenolic compounds and their antioxidant activity during two harvest seasons of palm fruit ethanol extract.

In my opinion, research into plant extracts and their compounds is very interesting and important in terms of their benefits and uses. In this paper, the authors have described the current status and outlined the aim of the work. I propose that the paper be accepted after the following corrections:

·         Throughout the article, write –trans, -cis, ortho-, -para, in vivo, in vitro,…............. in italics!

·         Write concentrations, mg/mL consistently throughout the article

·         Author et al. instead of Author and his collaborators throughout the article

·         Throughout the article: Fe2+  instead of Fe2+, IC50 instead of IC50, Trolox instead of trolox,

·         When comparing the results obtained with those of similar studies, indicate which extracts are involved.

·         Reagents and Chemicals: Edit this list by inserting a comma.

·         Plant material: Indicate who identified the plant material?

Table 2: Title: Content of individual phenolic compounds determined….(of palm fruits??)

·         Line 144: „…517 nm [61] and the ferric…“ instead   „.. 517 nm [61]. And the ferric…“

·         Line 579: "was driven by its non-toxic properties" Remove this part of the sentence

Author Response

Dear esteemed reviewer,

Thank you for taking the time to review our manuscript.

Please find the enclosed file.

Best regards,

Reviewer 2 Report

Comments and Suggestions for Authors

lines 91-94 of the introduction indicate rersults from the current work. I reccomend deleting

lines 100-104 do not need to be present

In tables, I suggest to reduce the number of digits, to make them more understandable.

Lines 362-392 this section must be rewritten, as it is confusing. Authors aim to compare obtained data with previous data, but sentences are long and confusing. 

I suggest removing tyrosol and quercetin from tables, as they were not detected

I reccomend another approch to statistical analysis. as it is, it makes comparision between years or cultivars difficult For instance, for Trans-ferulic acid, Khalt Khal in 2022 and Majhoul in 2022 are statistically different. But, if you analize data only for 2022 alone, i'm not quite sure that this will occur. 

Lines 410-433- another section a bit confusing. Authors can reduce these sentences, pointing out only key results

lines 440-464 these review on the properties of phenolics must be reduced. 

Authors provide a PCA aiming the evaluation of the correlation between various variables and to identify groupings within the samples. This is a good approach, but, actually, this PCA only accounts for 68% of the data. It use, for the dictintion of cultivars can be questionable. One appoach that can be performed by authors is the use of an LDA, to identify the most important variable. 

Furthermore, performing PCA for separate years provided similar results?

Authors refer to Figure 5, that I was unable to find in the manuscript (a figure 4 as well).

Can authors provide some insights on why the variation in the composition occured? Saying that it is link to genetic backround and climate is ok, but further explanations regarding this last item must be given. Was 2021 hotter that 2022?What about rain?

In the methodology, how much sample was collected? Please state the number of fruits (per replicate and total) or the weight.

Authors present results in dry weight. But I do not find any drying step on the methodology. Only storage at -20C, pitting and maceration. Please provide further details

Comments on the Quality of English Language

.

Author Response

(The authors gave the same response as above.)
